# Body Condition Score, Rumination, Intake, Milk Production and Milk Composition of Grazing Dairy Cows Supplemented with Rumen-Protected Lysine and Methionine

**Long Cheng** [1,*] **, Razaq Balogun** [2] **, Fanzeng Meng** [1] **, Frank R. Dunshea** [3,4] **and Brendan Cullen** [3]

1   Faculty of Veterinary and Agricultural Science, University of Melbourne, Dookie, VIC 3647, Australia; fanzengm@student.unimelb.edu.au
2   Kemin Industries (Asia) Pte Ltd., 12 Senoko Dr, Singapore 758200, Singapore; razbalogun@yahoo.com
3   Faculty of Veterinary and Agricultural Sciences, School of Agriculture and Food, The University of Melbourne, Parkville, VIC 3010, Australia; fdunshea@unimelb.edu.au (F.R.D.); bcullen@unimelb.edu.au (B.C.)
4   Faculty of Biological Sciences, The University of Leeds, Leeds LS2 9JT, UK
*   Correspondence: long.cheng@unimelb.edu.au; Tel.: +61-(0)-3-58339223

**Abstract:** The study utilised a pasture grazing based, voluntary traffic automatic milking system to investigate milk production of cows fed a pasture-based diet and supplemented with a pellet formulated with vs. without rumen-protected lysine and methionine (RPLM). The study adopted a switch-over design (over two periods of 5 and 10 weeks, respectively) and used 36 cows and equally allocated them into two experimental groups. The RPLM (Trial) pellet had 2% lower crude protein, but similar metabolizable energy content compared to the Control pellet. Pellet intake was 10.0 and 9.4 kg/day/cow. Milk yield was 36.2 and 34.4 kg/day/cow ($p = 0.23$), and energy corrected milk was 35.1 and 33.8 kg/day/cow ($p = 0.076$), and milk solids was 2.55 and 2.46 kg/cow/day ($p = 0.073$) in the Control and Trial groups, respectively. Milk fat%, milk protein%, milk fat: protein ratio, milking frequency and rumination time were not different between the two groups ($p > 0.05$). In period 1, plasma glucose was 3.1 mmol/L for both groups and milk urea were 150 and 127 mg/L in the Control and Trial groups, respectively. Both plasma glucose (as a proxy for energy supply) and milk urea (as a proxy for nitrogen use efficiency; NUE) were not different between groups ($p > 0.05$). This study showed that under a grazing pasture system, feeding lactating dairy cows a low protein pellet with RPLM supplementation, maintained milk production performance and NUE, compared with cows fed a high protein Control pellet diet with no RPLM. Further research should assess the long-term (seasonal) effects of feeding a diet formulated with RPLM on cow intake, health and reproductive performance.

**Keywords:** cattle; amino acid; robotic dairy; nitrogen efficiency



## 1. Introduction

Methionine (Met) and lysine (Lys) are often the first two limiting amino acids (AA) in the lactating dairy cow diet [1–3]. The effects of supplementing dairy cows with Met and Lys were previously investigated in total mixed ration (TMR) systems, particularly in European countries and United States [4,5]. In an earlier study, lactating dairy cows fed on a soybean meal-based diet supplemented with Met had a higher milk yield and milk protein yield than those ones fed with only soybean meal [6]. Schwab et al. [7] found that Lys and Met supplementation in lactating dairy cow diets stimulated feed intake, milk yield and milk protein%. However, other similarly designed studies showed contrasting results. There was limited or no improvement in feed intake and milk production performance in dairy cows supplemented with Met and Lys. For example, Rogers et al. [1] showed that Met and Lys supplementation had no influence on lactating dairy cows feed intake when a range of basal diets were offered, which included soybean meal, corn gluten meal and

urea-based diets. Further, it showed that lactating dairy cows fed on a soybean meal-based diet supplemented with Met and Lys had no change in their milk yield and milk protein yield compared with those cows fed with only soybean meal. In Ireland, Australia and New Zealand, dairy production systems are mainly based on grazing pastures, with TMR herds comprising a small proportion of the national dairy production (e.g., less than 3% of the Australian national herd [8]). To the best of our knowledge, limited literature can be found regarding the efficacy of rumen-protected AA (RPAA) in pasture grazing based dairy production system. Rulquin and Delaby [9] reported that when grazing lactating dairy cows were supplemented with rumen-protected Met, milk yield was slightly reduced (1.8%), with no changes in milk fat, fat yield and protein yield. The seasonal changes in the nutritive value of grazing pastures may well interacted with RPAA supplementation and altered dairy cow milk production and composition. Therefore, the first objective of this study was to investigate the effect of incorporating RPAA in the grazing pasture diet of dairy cows on milk production performance and body condition score (BCS) change.

Lactating dairy cows in fresh pasture based grazing system are often offered a diet containing a high ratio of rumen degradable to undegraded dietary protein relative to their requirement [10] and this can lead to low nitrogen use efficiency (NUE; milk nitrogen output/nitrogen intake) and high output of nitrogen into the environment, particularly in the form of urinary nitrogen excretion [10,11]. This can lead to substantial greenhouse gas emissions (e.g., nitrous oxide) to the atmosphere, which contributes to global warming and climate change. On the other hand, it can also cause nitrate leaching through soil profile, which can pollute ground water, promote algal growth and reduce fish population in rivers [12]. Previous work indicated that a reduction in crude protein (CP) may improve NUE [13], while maintaining or improving milk production performance of lactating dairy cows [14]. While few studies reported no effect on milk production when dietary CP was reduced [15,16], others reported increased milk production [17,18] and reduced milk protein% [19]. In such cases, supplementation with RPAA may be a useful strategy to balance AA in the diet of pasture based grazing dairy production system, which may help to alleviate the possible negative milk production effects of decreased dietary CP supply [20]. Therefore, the second objective of this study was to test if rumen-protected lysine and methionine (RPLM) supplementation with reduced CP supply will improve NUE of pasture grazing lactating dairy cows.

## 2. Material and Methods

### 2.1. Ethics Statement

The experiment was approved by the Animal Ethics Committee, The University of Melbourne, Melbourne, Victoria, Australia (Protocol no. 1814463.3).

### 2.2. Experimental Design and Feeding

The experiment was conducted at The University of Melbourne Dookie Dairy Farm (36°22′48″ S, 145°42′36″ E) from 1 June to 30 October 2018 (Winter/Spring). The farm consists of 41 ha of border check irrigated pastures, a feedpad and a milking herd of approximately 150 cows [6]. The farm has a split calving pattern, with approximately two-thirds of the cows calving in spring, and one-third in summer/autumn. Cows are milked in a Lely T4C Astronauts (The Netherlands) automatic milking system (AMS), with semi-voluntary cow movement through three pasture grazing areas to facilitate a target milking frequency of three times per day per cow. A total of 36 lactating dairy cows were used in the experiment. Eighteen cows were assigned to a pellet without RPLM (Control) and another 18 cows were assigned to a lower CP pellet, formulated with rumen-protected Lys and rumen-protected Met (LysiPEARL™ and MetiPEARL™, respectively; Kemin Industries, Inc., Des Moines, IA, USA) (Trial). The two dietary groups had similar stages of lactation, number of lactations, live weight, milk yield and milk solids production. The study was conducted over two periods; period 1 and 2, each comprised of one-week of feed adaptation, followed by a 5- and 10-week measurement periods, respectively. The

pellet offered were 13 and 11 kg DM/cow/day in period 1 and 2, respectively, through a calibrated automatic feeding system. Since the experiment adopted a switch-over design, upon the completion of period 1, a 4-week wash-out period was introduced prior to the start of period 2. Cows assigned to the Control and Trial pellets were swapped and allocated to the Trial and Control pellets in period 2, respectively. Cows had access to fresh pastures daily and silage, hay and straw were offered each day on the feedpad following milking, as per standard practice. The average monthly feed intake for the entire herd over the experimental period is shown in Table 1 and AA profile of feeds are shown in Tables S1 and S2. The milking herd size ranged from an average of 92 cows in July to 145 cows in October. Pellet (12.5 MJ metabolizable energy/kg DM) intake was recorded by the AMS, while the intakes of silage, hay and straw (average 10.5 MJ metabolizable energy/kg DM) were estimated by the amount offered and assuming at a wastage rate of 10%. A back-calculation technique was used to estimate pasture intake. The energy from pasture intake was estimated by first calculating the total animal energy requirements (using daily milk production and composition, body weight and gestation, following [21], then subtracting the energy supplied from pellet and silage/hay/straw.

**Table 1.** Pellet, silage/hay/straw and estimated pasture intake for the entire herd from June-October 2018.

| Month | Pellet Intake (Kg DM/Cow/Day) | Silage/Hay/Straw Intake (Kg DM/Cow/Day) | Pasture Intake (Kg DM/Cow/Day) |
|---|---|---|---|
| June 2018 | 7.7 | 10.5 | 2.0 |
| July 2018 | 7.1 | 9.5 | 2.9 |
| August 2018 | 7.1 | 2.6 | 8.4 |
| September 2018 | 5.4 | 0.7 | 10.5 |
| October 2018 | 5.5 | 0.3 | 14.3 |

### 2.3. Sampling and Measurements

Throughout the study, pellet and forage samples were collected fortnightly and weekly, respectively, for both nutritional value and AA analyses (per AOAC official method of 994.12). An experienced technician assessed the BCS of each cow at the start and end of each period. Daily, the robotic milking system recorded individual cow pellet intake, milk production, milking frequency and live weight. The robotic milking system also estimated milk composition (fat%, protein% and somatic cell count). Wearable Lely rumination collars (Qwes-HR; Lely Industries NV, Maasland, The Netherlands) were used to record rumination activity. Milk and plasma samples were collected in the last measurement day from individual cows in period 1 and analysed for milk urea content and blood glucose. Energy corrected milk (ECM; kg) was calculated as: (kg milk production $\times$ (0.383 $\times$ milk fat% + 0.242 $\times$ milk protein% + 0.7832) ÷ 3.1138) according to Østergaard et al. (2003) [22].

### 2.4. Data Analysis

The data collected for individual cow was averaged over each trial period and used in the one-way analysis of variance (ANOVA) with Genstat (version 16). Two statistical analysis methods were applied in the study. (1). treating individual animal as replicate in each treatment group per period (2). two-period data from the same dietary treatment was treated as replicates. For period 1 plasma glucose and milk urea data, they were analysed using individual cows from each group as the replicate unit in one-way ANOVA analysis. The overall result using (1) and (2) methods were similar; therefore, we only presented analysis method (2) data in the result section.

### 3. Results

The diet in experimental period 1 comprised pasture, wheat silage, cereal hay and pellet; while the diet in experimental period 2 comprised pasture, straw and pellet (Table 2). In both

experimental periods, the formulated pellet with RPLM had approximately 2% lower CP, but similar metabolizable energy (ME) content compared to the Control pellet (Table 2). The CP% (DM basis) of grazing pasture in period 1 was 14% higher than in period 2, and similar ME was observed across two periods in grazing pasture (11.9 vs. 11.2 MJ ME/kg DM).

**Table 2.** Nutritive value for forages and pellets.

| Diet Ingredient | DM (%) | Crude Protein (%; DM Basis) | Neutral Detergent Fiber (%; DM Basis) | Acid Detergent Fiber (%; DM Basis) | Metabolisable Energy (MJ/kg DM) |
|---|---|---|---|---|---|
| | | | Period 1 | | |
| Forages: | | | | | |
| Pasture | 18.9 | 23.8 | 38.1 | 22.3 | 11.9 |
| Wheat silage | 51.8 | 12.9 | 52.1 | 34.4 | 9.9 |
| Cereal hay | 87.8 | 5.8 | 54.1 | 38.1 | 9.9 |
| Pellet: | | | | | |
| Control | 87.1 | 16.9 | 17.8 | 3.6 | 12.1 |
| Trial | 88.1 | 15.1 | 18.1 | 3.7 | 12.3 |
| | | | Period 2 | | |
| Forages: | | | | | |
| Pasture | 20.0 | 20.9 | 41.1 | 23.3 | 11.2 |
| Straw | 90.3 | 4.1 | 78.3 | 56.8 | 6.9 |
| Pellets: | | | | | |
| Control | 89.0 | 17.4 | 17.7 | 3.8 | 12.5 |
| Trial | 89.3 | 15.4 | 17.2 | 3.8 | 12.8 |

There was no difference (*p* > 0.05) between control and treatment groups for pellet intake (10.0 vs. 9.4 kg/day/cow, respectively), milk yield (36.2 vs. 34.4 kg/day/cow, respectively) and ECM (35.1 vs. 33.8 kg/day/cow, respectively) (Table 3). Similarly, milk fat% and milk protein% were not different (*p* > 0.05) between the treatment groups (3.8% vs. 3.9%; 3.2% vs. 3.2% for Control and Trial group, respectively) (Table 3). There was no difference (*p* > 0.05) in milking frequency and rumination time (2.4 vs. 2.3 times/day/cow; 427 vs. 426 min/day/cow for Control and Trial group, respectively) as well as BCS change (0.03 vs. −0.01 unit) (Table 3). Milk fat: milk protein ratio and milk solid production were also similar across two groups (*p* > 0.05) (Table 3). In period 1, plasma glucose was 3.1 mmol/L for both groups and milk urea were 150 and 127 mg/L for Control and Trial group, respectively. Both plasma glucose and milk urea were not significantly different between groups (*p* > 0.05).

**Table 3.** The effect of amino acid supplementation on pellet intake, live weight, rumination time and milk production performance.

| Parameters | Control | Trial | Standard Error of Difference | *p* Value |
|---|---|---|---|---|
| Pellet intake (kg/cow/day) | 10.0 | 9.4 | 0.53 | 0.49 |
| Live weight (kg/cow) | 666 | 663 | 4.8 | 0.72 |
| Body condition score change | 0.03 | −0.01 | 0.126 | 0.79 |
| Rumination (min/cow/day) | 427 | 426 | 26.7 | 0.98 |
| Milk yield (kg/cow/day) | 36.2 | 34.4 | 0.68 | 0.23 |
| Milk solids (kg/cow/day) | 2.55 | 2.46 | 0.011 | 0.073 |
| Milking frequency (times/cow/day) | 2.44 | 2.34 | 0.024 | 0.14 |

**Table 3.** *Cont.*

| Parameters | Control | Trial | Standard Error of Difference | p Value |
|---|---|---|---|---|
| Milk fat (%) | 3.82 | 3.93 | 0.037 | 0.20 |
| Milk protein (%) | 3.25 | 3.24 | 0.136 | 0.98 |
| Milk fat: protein ratio (g/g) | 1.17 | 1.21 | 0.042 | 0.53 |
| Milk somatic cell count (×1000 cells/mL) | 113 | 115 | 13.4 | 0.92 |
| Energy corrected milk (kg/cow/day) | 35.1 | 33.8 | 0.16 | 0.076 |
| Plasma glucose (mmol/L) from period 1 only | 3.09 | 3.14 | 0.106 | 0.64 |
| Milk urea (mg/L) from period 1 only | 150 | 127 | 20.1 | 0.27 |

## 4. Discussion

Rumination activity was not different between the two groups, with an average rumination of 426 min/day/cow observed in this study, similar to that observed previously in other studies (453–465 min/cow/day from [23] and 357–618 min/cow/day from [24]). The plasma glucose concentration is a proxy for energy supply in lactating dairy cow body and milk fat: protein ratio is generally used as an indicator for energy deficit [25] or subclinical ketosis [26]. The results between the two groups indicated that cows from both groups had similar energy supply from the diet and energy metabolism status. As energy supply is closely related to milk production, this may attribute to the major finding from this study, that is, no significant influence of a low CP diet with Lys and Met supplementation on production performance of lactating dairy cows under this study conditions. This is consistent with results found in previous work that investigated RPAA supplementation in a TMR feeding system [14].

As the cows were milked through a voluntary traffic automatic milking system (i.e., the voluntary movement of cattle around a farm) in this study, the milking frequency may impact on milk production performance from each group [27]. Therefore, milk production comparison between two groups may be adjusted for milking frequency (2.44 vs. 2.34 times/cow/day). The adjusted results demonstrated no difference in milk production performance between the Control and Trial groups (data not shown). Further analysis to explore energy corrected milk production (kg/cow/day), which determines the amount of energy in the milk, showed no significant difference between the two groups.

In terms of milk composition, no difference between two groups was found for milk somatic cell count in this study. Rulquin and Delaby (1997) reported that when grazing cows were supplemented with Met, no changes in milk fat% and fat yield were found [9], which agrees with our findings. Similarly, Rogers et al. (1989) showed that cows supplemented with Met and Lys exhibited no change in milk protein yield compared to cows fed without supplementation [1].

Milk urea or milk urea nitrogen content is a well-established biomarker for NUE in lactating dairy cows [13]. We found no difference in milk urea content between the two groups. This implies that there was unlikely to be any difference in NUE between dietary treatments in this experiment. In this current study, cows fed with Trial pellet had similar BCS to the Control pellet group, indicating that incorporating RPLM into a diet with reduced CP was unlikely to negatively impact cow body reserve (e.g., mobilise body fat to support milk production) during the lactation. It is important to note that the direct measure of cow NUE was not performed in this study, as it involves measuring individual cows CP intake and milk protein yield [13]. Therefore, future work with direct measures of NUE is needed to confirm our findings. Further research should assess the

long-term (seasonal) effects of feeding a diet formulated with RPAA on cow intake, health and reproductive performance.

## 5. Conclusions

This study showed that under a pasture grazing based system, feeding lactating dairy cows a reduced CP diet fortified with RPLM maintained milk production performance and NUE, compared with cows fed a control diet with no RPLM supplementation. Further research should assess the long-term (seasonal) effects of feeding the reformulated pellet on cow intake, health and reproductive performance.

**Supplementary Materials:** The following are available online at https://www.mdpi.com/article/10.3390/dairy2030037/s1, Table S1: Amino acids analysis of pellet and forage (Period 1), Table S2: Amino acids analysis of pellet and forage (Period 2).

**Author Contributions:** Conceptualization, L.C., R.B., F.R.D. and B.C.; methodology, L.C., R.B., F.R.D. and B.C.; formal analysis, L.C., R.B. and F.M.; writing—original draft preparation, L.C., R.B., F.M. and B.C.; writing—review and editing, L.C., R.B., F.M., F.R.D. and B.C.; project administration, L.C. and R.B.; funding acquisition, L.C. and R.B. All authors have read and agreed to the published version of the manuscript.

**Funding:** This research was funded by Kemin Industries (Asia) Pte Ltd.

**Institutional Review Board Statement:** All methods were carried out in accordance with the relevant guidelines and regulations, and all animal manipulations were approved by The University of Melbourne (Australia).

**Informed Consent Statement:** Not applicable.

**Data Availability Statement:** All the raw data regarding this work are available from L. Cheng.

**Acknowledgments:** We appreciate Kemin Industries (Asia) Pte Ltd. (Singapore) for the financial support. Special thanks to Dookie Dairy farm manager, Damien Finnigan.

**Conflicts of Interest:** We declare that co-author Dr. Razaq Balogun was an employee at Kemin Industries (Asia) Pte Ltd. when the study was conducted. Dr. Balogun supported objective and valuable contributions to the design of the study and has proof-read the manuscript prior to submission. Kemin Industries (Asia) Pte Ltd. funded this project.

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
