# Peer review of "Body Condition Score, Rumination, Intake, Milk Production and Milk Composition of Grazing Dairy Cows Supplemented with Rumen-Protected Lysine and Methionine"

_2624-862X, doi:10.3390/dairy2030037_

Round 1

Reviewer 1 Report

Dear authors,

  • I have seen some comments in the work, I hope that such comments will be taken into account. I want to shed light on these points:
  • In line 58 it is better to write feed intake instead of just feed.
  • In line 56, the reference number must be placed directly after the name and not the date of publication, as the publication date is given in the reference list. That is, Schwab et al. (7) and not Schwab et al. (1993). This matter needs to be taken into account with other references as like in the lines: 62, 65, 75.
  • Between lines 62 and 68 it is better if this part is written again, as some information is repeated.
  • In line 119, which means the number 6 between brackets.
  • On Line 146 is an abbreviation AMS that was not explained before.
  • In lines 152 and 153 a reference was written that has no number and is not in the reference list.
  • In lines 187 to 195, the authors have given information that must be written in material and methods. The table number (2) must also be written in Material and Methods.
  • The results show that there were no significant differences between the two groups in terms of the parameters tested, as the study duration was short. In my opinion, if the study period is longer, there is a higher probability that there are significant differences between the two groups.

I wish you success

Author Response

  • In line 58 it is better to write feed intake instead of just feed.

Corrected in L 58

  • In line 56, the reference number must be placed directly after the name and not the date of publication, as the publication date is given in the reference list. That is, Schwab et al. (7) and not Schwab et al. (1993). This matter needs to be taken into account with other references as like in the lines: 62, 65, 75.

Corrected in L56, 62, 75, deleted L65 reference

  • Between lines 62 and 68 it is better if this part is written again, as some information is repeated.

Re written L62-68

  • In line 119, which means the number 6 between brackets.

It is the reference we cited to provide general farm description.

  • On Line 146 is an abbreviation AMS that was not explained before.

Added in L122

  • In lines 152 and 153 a reference was written that has no number and is not in the reference list.

Added in: Primary Industries Standing Committee. Nutrient Requirements of Domesticated Ruminants; CSIRO Publishing: Melbourne, Australia, 2007.

  • In lines 187 to 195, the authors have given information that must be written in material and methods. The table number (2) must also be written in Material and Methods.

We disagree this with referee, we published more than 80 journal paper and many of them described feed quality in the beginning of the result section.

  • The results show that there were no significant differences between the two groups in terms of the parameters tested, as the study duration was short. In my opinion, if the study period is longer, there is a higher probability that there are significant differences between the two groups.

We do not think the experimental duration in this case will make a big difference, as milk production response to diet treatment is quick (a few days). However, due to seasonal variation of pasture quality/conserved forage quality, it may interact with the supplementation of AA, which in turn effect on milk production.  

Reviewer 2 Report

This study showed that under a grazing pasture system, feeding lactating dairy cows a low protein pellet with RPLM supplementation, maintained milk production performance and NUE, compared with cows fed a high protein Control pellet diet with no RPLM.
If you say this is the conclusion of this study, what is the advantage of feeding a low protein pellet with RPLM ?
Have you compared the cost and the effort of buying and feeding extra feed by farmers ?
Do you think that feeding less CP and more Rumen Protected will produce less ammonia in the rumen and affect reproductive performance ?

L110-111; Why is the font style different only here?
L120; Lely T4C Astronauts --- Company place ?
L145; AMS --- What is this ?
L244; numerically 4% higher --- If you can say so, why did you conduct statistical analysis ? Why don't you simply compare only the size of the numbers? Why did you ignore the statistical analysis only here?
Table 3. S. E. D --- S. E. D. What is this ?

Author Response

This study showed that under a grazing pasture system, feeding lactating dairy cows a low protein pellet with RPLM supplementation, maintained milk production performance and NUE, compared with cows fed a high protein Control pellet diet with no RPLM.
If you say this is the conclusion of this study, what is the advantage of feeding a low protein pellet with RPLM ?

  1. Potentially reduce nitrogen excretion – an environmental benefit.
  2. Has the potential to reduce cost associated with high protein ingredients.

Have you compared the cost and the effort of buying and feeding extra feed by farmers ?

No.

Do you think that feeding less CP and more Rumen Protected will produce less ammonia in the rumen and affect reproductive performance ?

May be, lower NH3 may also reduce nitrogen excretion in the urine which reduces pollution.

L110-111; Why is the font style different only here?

Revised L110-111

L120; Lely T4C Astronauts --- Company place ?

L122 added in (Netherlands)

L145; AMS --- What is this ?

Added in L122

L244; numerically 4% higher --- If you can say so, why did you conduct statistical analysis ? Why don't you simply compare only the size of the numbers? Why did you ignore the statistical analysis only here?

We deleted this sentence.

Table 3. S. E. D --- S. E. D. What is this ?

Standard error of difference, spelled it out in Table 3